# Hsp22 with an N-Terminal Domain Truncation Mediates a Reduction in Tau Protein Levels

**DOI:** 10.3390/ijms21155442

**Published:** 2020-07-30

**Authors:** Jack M. Webster, April L. Darling, Taylor A. Sanders, Danielle M. Blazier, Yamile Vidal-Aguiar, David Beaulieu-Abdelahad, Drew G. Plemmons, Shannon E. Hill, Vladimir N. Uversky, Paula C. Bickford, Chad A. Dickey, Laura J. Blair

**Affiliations:** 1Department of Molecular Medicine, Morsani College of Medicine, University of South Florida, Tampa, FL 33620, USA; jackwebster@usf.edu (J.M.W.); April.Darling@Pennmedicine.Upenn.edu (A.L.D.); tsanders2@mail.usf.edu (T.A.S.); dblazier@usf.edu (D.M.B.); yvidalaguiar@gmail.com (Y.V.-A.); davidbeaulie@usf.edu (D.B.-A.); plemmonsd@usf.edu (D.G.P.); sehill2@usf.edu (S.E.H.); vuversky@usf.edu (V.N.U.); cdickey@health.usf.edu (C.A.D.); 2USF Health Byrd Alzheimer’s Institute, University of South Florida, Tampa, FL 33620, USA; 3Research Service, James A Haley Veterans Hospital, 13000 Bruce B Downs Blvd, Tampa, FL 33612, USA; pbickfor@usf.edu; 4Department of Neurosurgery and Brain Repair, Morsani College of Medicine, University of South Florida Health, 12901 Bruce B Downs Blvd, Tampa, FL 33612, USA; 5Department of Molecular Pharmacology and Physiology, Morsani College of Medicine, University of South Florida Health, 12901 Bruce B Downs Blvd, Tampa, FL 33612, USA

**Keywords:** small heat shock protein 22, tau, molecular chaperones, neurodegeneration, Alzheimer’s disease

## Abstract

Misfolding, aggregation and accumulation of proteins are toxic elements in the progression of a broad range of neurodegenerative diseases. Molecular chaperones enable a cellular defense by reducing or compartmentalizing these insults. Small heat shock proteins (sHsps) engage proteins early in the process of misfolding and can facilitate their proper folding or refolding, sequestration, or clearance. Here, we evaluate the effects of the sHsp Hsp22, as well as a pseudophosphorylated mutant and an N-terminal domain deletion (NTDΔ) variant on tau aggregation in vitro and tau accumulation and aggregation in cultured cells. Hsp22 wild-type (WT) protein had a significant inhibitory effect on heparin-induced aggregation in vitro and the pseudophosphorylated mutant Hsp22 demonstrated a similar effect. When co-expressed in a cell culture model with tau, these Hsp22 constructs significantly reduced soluble tau protein levels when transfected at a high ratio relative to tau. However, the Hsp22 NTDΔ protein drastically reduced the soluble protein expression levels of both tau WT and tau P301L/S320F even at lower transfection ratios, which resulted in a correlative reduction of the triton-insoluble tau P301L/S320F aggregates.

## 1. Introduction

Maintenance of cellular proteostasis is critically important for cellular function and survival [1,2,3], especially in neurons [4,5,6]. Misfolding, aggregation and disrupted protein clearance contribute to an imbalance in proteostasis that can foster the accumulation of toxic protein aggregates [6,7,8]. Aberrant accumulation of aggregated protein is often associated with neurodegenerative disease progression [9,10,11]. Therefore, strategies that leverage either a direct inhibition of protein aggregation or modulation of protein clearance mechanisms may have therapeutic potential [1,8,12,13]. Molecular chaperones can counteract this proteostatic imbalance by facilitating proper folding or refolding, sequestration, or clearance of misfolded proteins [14,15,16,17,18]. Small heat shock proteins (sHsps) are a class of ATP-independent molecular chaperones that associate with early misfolded proteins and sequester these aggregation-prone intermediates for processing by ATP-dependent molecular chaperone complexes that include the 70 kDa heat shock protein (Hsp70), the 90 kDa heat shock protein (Hsp90) [19,20], or other protein clearance machinery. sHsps canonically contain a conserved core α-crystallin domain (ACD) flanked by variable flexible N-terminal and C-terminal domains (NTD and CTD) [21,22]. sHsps form dimers through an interface between ACDs. sHsps further multimerize to form higher order dynamic oligomers via interactions of an IXI/V sequence in the CTD with a hydrophobic groove formed by the β4 and β8 strands of the ACD, as well as through complex interactions of the NTD with neighboring sHsp subunits [23,24].

In vitro, certain sHsps appear to have chaperone activity, reducing or delaying the aggregation of discrete amyloidogenic clients [19,23,25,26,27,28]. The aggregation of one of these client proteins, the microtubule-associated protein tau (tau), is thought to contribute to disease progression in multiple neurodegenerative diseases, including Alzheimer’s disease (AD) and frontotemporal dementia (FTD) [29,30]. The sHsp, Hsp27, interacts directly with tau and reduces tau aggregation in vitro [23,26,27,28] as well as neuronal tau accumulation in vivo [26]. We previously demonstrated that Hsp27 reduces neuronal tau accumulation and rescues a hippocampal tau-induced deficit in long-term potentiation (LTP), a measure of hippocampal plasticity [26]. However, overexpression of a pseudophosphorylated mutant (Hsp27 S15D/S78D/S82D) results in increased tau accumulation and a failure to rescue the hippocampal plasticity deficit, suggesting a requirement for these N-terminal residues to cycle between phosphorylated and unphosphorylated states. Freilich et al., demonstrated that while the ACD and NTD of Hsp27 are each capable to bind tau on their own, self-interactions of the NTD appear to obscure tau binding in vitro. However, the NTD was crucial for the ability of Hsp27 to delay tau aggregation [23]. Phosphorylation of the Hsp27 NTD results in a shift to smaller Hsp27 oligomers, which may additionally free up binding sites in the NTD and ACD to interact with tau.

Hsp22 is a sHsp that lacks the IXI/V sequence commonly found in CTDs, resulting in oligomeric assemblies, which are smaller than other sHsps [23,24,25]. However, the β4/β8 groove of the Hsp22 ACD may still interact with other sHsp family members as well as other proteins that contain an IXI/V interaction sequence, like the Bcl-2-associated athanogene protein-3, Bag3 [31]. Hsp22 promotes the autophagic clearance of many client proteins through an association with a multiprotein complex that includes Bag3, Hsp70 and C-terminus of Hsc70-interacting protein (CHIP) [31,32,33,34,35]. Like Hsp27, Hsp22 contains NTD phosphorylation sites that appear to regulate the self-oligomerization state and chaperone activity for selected substrates [36]. While a physical interaction between Hsp22 and tau has not yet been reported, Hsp22 did increase the lag time of in vitro heparin-induced aggregation for particular tau variants, including tau phosphomimetics T153E, S356E and S404E [28]. Additionally, Bag3 overexpression in neurons reduced tau aggregation [37], suggesting that autophagy may play a role in tau clearance and may further implicate a role for the Bag3-binding partner Hsp22 [38]. Here, we evaluated the effects of Hsp22, a non-phosphorylatable mutant (S24A/S57A), a pseudophosphorylated mutant (S24D/S57D) and an N-terminal domain deletion (NTDΔ) mutant on tau aggregation and accumulation in vitro and in a cellular model. While Hsp22 did demonstrate inhibition of tau aggregation in vitro, the effects of Hsp22 variants with mutations at phosphorylation sites S24 and S57 were not significantly different than Hsp22 WT. In a cellular model, Hsp22 and variants with point mutations at phosphorylation sites did not significantly affect tau levels. However, co-expression of Hsp22 with an NTD deletion (NTDΔ) demonstrated a robust reduction of both WT and mutant tau P301L/S320F expression, accompanied by a reduction in triton-insoluble aggregates.

## 2. Results

### 2.1. Hsp22 Expression with Age and in AD

We first sought to examine Hsp22 mRNA expression profiles in the human brain to determine whether expression levels are modulated with aging or in AD. Figure 1A shows that Hsp22 mRNA expression is increased in AD relative to young controls in superior frontal gyrus, hippocampus and entorhinal cortex, but not in posterior cingulate gyrus tissue. A significant difference between AD and age matched controls was only found in the entorhinal cortex, and a significant difference between young and aged control tissue was only found in the hippocampus. Specific increases may represent a cellular stress response to intracellular protein misfolding and accumulation. Additionally, we evaluated Hsp22 protein levels in human medial temporal gyrus from AD and age-matched control brain tissue by immunohistochemistry (Figure 1B,C). Hsp22 protein levels varied between cognitively normal aged patients, with two patients having high Hsp22 levels and two patients with low Hsp22 levels. All four AD patient tissues demonstrated relatively lower Hsp22 protein levels. Additionally, Hsp22 protein levels were evaluated in control (nTg) and transgenic P301L tau mice (rTg4510). Figure 1D,E shows Hsp22 staining in the hippocampi of the transgenic model was not significantly different than the control hippocampi.

### 2.2. Hsp22 Prevents the Formation of Tau Aggregates In Vitro

We then sought to evaluate effects of Hsp22 and variants on tau aggregation and cellular clearance. The NTD of sHsps is often flexible and disordered, making it susceptible to post-translational modifications, including phosphorylation. Phosphorylation of sHsps in the NTD has been demonstrated to regulate self-oligomerization and chaperone activity towards certain clients [39,40]. To investigate the direct effects of Hsp22 on tau aggregation, we purified Hsp22 WT and two variants with point mutations (Figure 2A). The first variant contains S24D and S57D point mutations to mimic a perpetually phosphorylated state (Hsp22 S/D). The second variant contains S24A and S57A point mutations, as a control variant with non-polar mutations (Hsp22 S/A). Figure 2B shows the predicted intrinsic disorder profile of Hsp22 and these two variants, which demonstrated a high propensity for disorder in the NTD. The disorder propensity of Hsp22 is only minimally affected by S24A/S57A and S24D/S57D point mutations that are causing local changes in disorder predisposition. In an effort to evaluate the effects of Hsp22 on tau, including FTD-related mutant forms with enhanced aggregation properties, we purified three different variants of 0N4R tau protein: tau WT, tau P301L, and tau ΔK280. To investigate whether tau aggregation could be altered by Hsp22, we conducted in vitro thioflavin T (ThT) assays. Since the Hsp22 S/D pseudophosphorylation variant was reported to have decreased chaperone activity for two substrates, insulin and rhodanese [36], we hypothesized that the Hsp22 S/D mutant might demonstrate reduced chaperone activity for tau. Results showed that all three variants of Hsp22 were able to reduce heparin-induced aggregation of each of the tau variants tested (Figure 3A); although Hsp22 S/D effects on tau P301L aggregation were not statistically significant relative to the control with no Hsp22 variant.

We further characterized the final ThT assay aggregated protein products by visualizing them using transmission electron microscopy (TEM). Results showed that each tau variant formed large fibrillar structures, and all three variants of Hsp22 were able to dramatically reduce the size of those fibrils. It should be noted that in the Hsp22 conditions, tau fibrils were still present but the size was reduced and appeared thinner (Figure 3B). Taken together, these data indicate that Hsp22 can directly prevent the aggregation of tau WT as well as FTD-related mutants tau P301L and tau ΔK280.

### 2.3. Hsp22 NTDΔ Exacerbates Tau P301L Aggregate Formation In Vitro

Since the effect of Hsp22 N-terminal pseudophosphorylation mutants on tau aggregation was not statistically different than that of Hsp22 WT (Figure 3A), we sought to evaluate the importance of the NTD using a truncated mutant (Hsp22 NTDΔ) as shown in Figure 2A. We performed ThT assays using tau P301L with different concentrations of Hsp22 WT or Hsp22 NTDΔ. Tau P301L aggregation was reduced with increasing concentrations of Hsp22 WT (lower tau: Hsp22 WT ratio); remarkably, an increase in aggregation measured by ThT was detected with increasing Hsp22 NTDΔ concentrations (Figure 4A). Additionally, increasing concentrations of Hsp22 NTDΔ shortened the lag phase as well as increased the rate of fibrillization detected by ThT. TEM visualization of the end product shows increased aggregated protein in the presence of Hsp22 NTDΔ, as shown in Figure 4B. Coomassie stained gels of purified recombinant protein and estimations of purity are shown in Appendix A. Melting temperatures for recombinant Hsp22 WT and Hsp22 NTDΔ were estimated using differential scanning fluorimetry (DSF), showing that truncation of the NTD does not decrease the protein stability (Figure 4C).

### 2.4. Hsp22 NTDΔ Enhances the Clearance of Tau Protein in HEK293T Cells

To determine the effects of Hsp22 on tau in a cellular environment, HEK293T cells were co-transfected with either a tau WT or tau P301L/S320F expression plasmid and either an empty vector or an Hsp22 variant expression plasmid as indicated (at a 1:5 tau vector:Hsp22 vector ratio). In the triton-soluble fractions (Figure 5A,B, left panels), there was a slight but not significant reduction of tau WT levels in the presence of Hsp22 WT as well as the Hsp22 S/A mutant, yet only Hsp22 NTDΔ expression resulted in a significant reduction of tau WT protein. Similarly, there was a trend toward reduction of tau P301L/S320F levels in the presence of Hsp22 WT and Hsp22 S/A, which was not observed with the Hsp22 S/D variant. Only Hsp22 NTDΔ expression resulted in a significant reduction of tau P301L/S320F protein. Tau P301L/S320F was chosen for the cell assay because it forms triton-insoluble aggregates when expressed in HEK293T cells [41], therefore triton-insoluble fractions were also evaluated for tau content (Figure 5A,B, right panels). As expected, tau WT protein was not detected in the insoluble fractions. Differences in tau P301L/S320F content in the insoluble fraction followed the same pattern of tau P301L/S320F levels with co-expressed Hsp22 variants in the triton-soluble fraction, indicating that a reduction of aggregation was likely due to a reduction in total tau protein and could not be directly attributed to an effect of the chaperones on preventing tau aggregation. It is also notable that there appear to be two tau immunoreactive bands present, with the lower band more prominent in the triton-soluble fractions and the higher band more prominent in the triton-insoluble fractions. Total protein yield in each soluble fraction was measured and shown in Appendix A, indicating that all cotransfection conditions did not result in gross cellular toxicity or impaired cell growth. GAPDH immunoblots and Ponceau S staining of PVDF membranes after transfer indicate equivalent loading of protein in each sample as well as efficient transfer (Figure 5A and Appendix A).

Triton-soluble fractions were also probed for Hsp22 expression (Figure 5A, middle panel). While Hsp22 WT and point mutation variants were expressed as expected, a lower molecular weight Hsp22 NTDΔ was initially not detected. However, the transfection of the Hsp22 NTDΔ expression plasmid had the most robust effect on tau expression or clearance, which strongly suggested this deletion variant was indeed expressed. Analysis of more concentrated triton-soluble fractions revealed that the Hsp22 NTDΔ protein was indeed produced and immunoreactive to an antibody directed to the CTD, albeit at a much lower level than the other Hsp22 variants (Appendix A). Additionally, to ensure that the low expression of Hsp22 NTDΔ in this HEK293T cell model is not dependent on the co-expression of tau protein, single transfections of Hsp22 WT and Hsp22 NTDΔ plasmids were compared demonstrating a similar impaired expression in the deletion mutant (Appendix A).

### 2.5. Hsp22 WT and Phosphorylation Site Mutants Enhance the Clearance of Tau Protein in HEK293T Cells at High sHsp Ratios

Next, we performed experiments similar to those shown in Figure 5, but with a 1:10 ratio of tau vector to Hsp22 vector, which showed a significant reduction in soluble tau WT levels in response to Hsp22 WT, Hsp22 S/A, HSP22 S/D and Hsp22 NTDΔ (Figure 6A,B). Tau P301L/S320F protein levels were also reduced in the presence of all the Hsp22 variants but were only significantly different from empty vector controls with Hsp22 S/A and Hsp22 NTDΔ. Insoluble tau P301L/S320F levels appeared to be reduced with Hsp22 NTDΔ co-expression but the results were not statistically significant.

## 3. Discussion

Previous work has shown that certain molecular chaperones, including a sHsp (Hsp27), can interact with tau protein and inhibit or delay aggregation [26,28]. The NTD of sHsps is often a flexible disordered region that is susceptible to post-translational modifications, including phosphorylation. Phosphorylation of many sHsp NTDs appears to affect the formation and dynamic cycling between higher order oligomers, which is often correlated to chaperone activity [39,40]. Here, we evaluated the effects of Hsp22 WT, a pseudophosphorylated mutant (S/D), and a mutation with non-polar residues (S/A) on tau aggregation in vitro. While they all appeared to reduce in vitro tau aggregation, we did not find evidence for any differences in modulation of tau aggregation between Hsp22 WT and Hsp22 S/A or Hsp22 S/D. We hypothesized that one or more of the Hsp22 variants would not affect in vitro tau aggregation and serve as a negative control; since all Hsp22 variants modulated tau aggregation we cannot rule out a non-specific effect of additional bulk protein in the assay. This differs from the effect of Hsp27 on in vitro tau aggregation, as our group reported less robust inhibition of tau aggregation for a pseudophosphorylated Hsp27 variant [26]. This suggests that the NTD of Hsp22 may have a different role than the NTD of Hsp27 with respect to the tau client. Indeed, as Hsp22 oligomers are much smaller than Hsp27 oligomers, the role of NTD phosphorylation dynamics may play a smaller role in changes to sHsp oligomer size and chaperone activity. The Hsp27 NTD appears to enable the formation of large oligomers, which may function to obscure hydrophobic regions of both the NTD and ACD in order to prevent its own aberrant aggregation, which can also functionally obscure tau binding sites. It has been suggested that in the case of Hsp27, binding to misfolded tau requires tau-sHsp interactions that compete for these oligomer-forming binding sites [23]. Therefore, it is not surprising that in vitro tau aggregation in the presence of Hsp22 NTDΔ resulted in an increase in the formation of β-sheet containing aggregates. This effect may be due to hydrophobic regions of the Hsp22 NTDΔ that contribute to early oligomer/seed formation as well as incorporation as a constituent of the forming β-sheet-rich aggregates in this assay. This proposed in vitro mechanism is supported by the shortened lag phase, increased rate of fibrillization, and the increased total yield of fibrillized protein in the presence of Hsp22 NTDΔ, seen in Figure 4. This mechanism contrasts with Hsp22 WT, which reduced the total yield of fibrillized protein by reducing the rate of aggregation through the elongation phase. This suggests that the effects of Hsp22 WT on tau aggregation may occur at a mid to late stage of fibril elongation, while not directly impacting early tau oligomer, or seed, formation. This finding is consistent with published data demonstrating no effect on tau P301L aggregation lag time by Hsp22 or Hsp27, and an effect of Hsp27 to lengthen the lag time of tau WT (0N4R) aggregation to a greater degree than Hsp22 [23,28].

The most disordered part of Hsp22 is its N-terminal domain (residues 1-71), whereas in Hsp27, disorder is preferentially concentrated within the C-terminal region of the protein (residues 150-205) [42]. Hsp27 chaperone activity for tau appears to require the NTD and is sensitive to changes in phosphorylation [23,26]; here, we demonstrate that the ability of Hsp22 to mediate a reduction in tau expression levels does not require the NTD. The removal of the mostly disordered N-terminal domain, NTDΔ, resulted in an Hsp22 variant with a mean disorder score of 0.43 and 26.8% disordered residues, which is noticeably more ordered than Hsp22 WT (Figure 2B).

While the robust effect of Hsp22 WT to inhibit tau aggregation in vitro suggests a potential role for direct chaperone activity of Hsp22 complexes to reduce tau aggregation, our cell co-transfection model did not provide evidence for a direct inhibition of tau aggregation. We utilized expression of tau P301L/S320F, a combination of two FTD mutations, as an accelerated model of cellular tau aggregation [41], which demonstrated triton-insoluble aggregates that were not present with tau WT. While there were reductions in tau P301L/S320F levels in the insoluble fractions with Hsp22 variant co-expression, they correlated well with a decrease in tau P301L/S320F levels in the triton-soluble fraction. This suggests that the reduction in aggregated tau may be due more to clearance of total tau and less to direct steric inhibition of aggregation attributed to tau-binding that we find in vitro. While only Hsp22 NTDΔ significantly reduced tau levels in our standard experiment using plasmid vectors at a 1:5 ratio of tau to sHsp, further increasing the ratio of sHsp to tau vectors resulted in a significant reduction in tau WT levels for all of the Hsp22 variants examined. Overall, this suggests that Hsp22 WT, Hsp22 S/A Hsp22 S/D, and Hsp22 NTDΔ are each capable of enhancing the clearance of tau in cells; however, Hsp22 NTDΔ has a more potent effect on cellular tau clearance. This suggests a potential regulatory role of the Hsp22 NTD toward the client tau that is independent of phosphorylation at serines 24 and 57.

While Hsp22 NTDΔ resulted in exacerbated aggregation in vitro, a very different result was demonstrated in our cell culture model. The robust effect of Hsp22 NTDΔ to reduce protein levels of exogenous tau WT and tau P301L/S320F may be due to the loss of competitive inhibition to tau binding afforded by the NTD. This may result in an Hsp22 variant with increased affinity (or availability) for the intrinsically disordered tau protein. In a cellular context, exogenous Hsp22 variants likely partner with additional endogenous proteins and cellular clearance components that are absent in our in vitro aggregation assay. Since Hsp22, in complex with proteins like Bag3, plays a role in segregating clients for clearance by autophagy, a reduction in protein expression levels of client proteins, like tau, may be expected. Future studies utilizing overexpression or knock down of Hsp22-interacting proteins, like Bag3, or proteins that competitively inhibit Hsp22 interactions with Hsp70/CHIP complexes, like Bag1, may allow the regulation of a switch between an Hsp70/CHIP/Bag1 complex driving proteasomal clearance of certain clients to an Hsp70/CHIP/Bag3/Hsp22 complex driving autophagic degradation of clients [43] like tau. The mechanism of tau protein reduction by Hsp22 NTDΔ has not been determined, but further evaluation in the presence of autophagy or proteasome inhibitors may distinguish the mechanism of tau clearance. Additionally, comparison of the effects of NTD variants or deletions on tau protein levels of other sHsps, like Hsp27, may provide insights into the role of sHsps on tau aggregation.

Hsp22 mRNA expression was upregulated in human brain regions with aging and in AD (Figure 1A). However, differences in Hsp22 mRNA between age-matched controls and AD were only statistically significant in the entorhinal cortex. Protein levels from AD and age-matched controls in another brain region, human medial temporal gyrus, demonstrated wide variability in aged brain and generally lower levels showing variability in expression among AD patients. The increase in Hsp22 mRNA expression in aging and AD may be a regulatory mechanism to address deficiencies in proteostasis and the accumulation of misfolded proteins, but we do not see an obvious upregulation of Hsp22 protein levels in brain tissue from AD patients or a mouse model of tauopathy. It is possible that Hsp22 gene expression does not lead to observable increases in Hsp22 protein due to enhanced turnover of the functioning chaperone. In fact, recombinant Hsp22 protein is unique among sHsps in that it is susceptible to proteolytic degradation when incubated in HEK293T cell lysates at 45 °C, even in the presence of protease inhibitors [25]; therefore this protein appears to be more sensitive to post-translational regulation. Additional studies are needed to understand the dynamics of cell type specific Hsp22 expression and regulation. One recent study identified lower Hsp22 expression in excitatory neurons compared to inhibitory neurons [44]. This is important because this same study showed that excitatory neurons are more sensitive to tau toxicity. Thus, it is possible that Hsp22 levels are not properly being upregulated in neurons that degenerate. There may be an opportunity for designing engineered sHsp variants with high chaperone activity towards tau and/or resistance to proteolytic degradation.

Modulating the expression or activity of endogenous molecular chaperones is an intriguing strategy to reduce aggregation of amyloidogenic proteins like tau in vivo. Additionally, gene therapy vectors allow the exogenous expression of molecular chaperones in the brain, as we have demonstrated with Hsp27, Aha1 and Cyp40 in an animal model of tauopathy [14,15,26]. Utilizing exogenous chaperones will allow the incorporation of potentially beneficial mutations, engineered proteins, or fusion proteins. Continued evaluation of the mechanism by which Hsp22 NTDΔ reduces cellular tau expression and aggregation in cell models and in animal models of neurodegeneration may enhance our understanding of the role of Hsp22 in the clearance of aggregation-prone proteins and provide guidance in the development of engineered molecular chaperones as therapeutics. It is remarkable that Hsp22 NTDΔ variant has such a robust effect on tau, while demonstrating markedly reduced expression relative to Hsp22 WT and other Hsp22 variants. It is possible that Hsp22 NTDΔ itself is recognized as misfolded and cleared, which could explain the lower protein levels (an apparent shorter half-life) of this non-natural protein relative to Hsp22 WT, shown in Appendix A. Such a mechanism would alleviate potentially toxic effects that might result from potential self-aggregation of Hsp22 NTDΔ. The enhanced turnover of this chaperone fragment could also facilitate clearance of associated client proteins via the same mechanism. This mechanism may be akin to engineered antibody fragments fused with degron signal peptides that facilitate their own degradation as well as their target protein [45,46]. It is also notable that the reduction in tau protein was evident for both tau WT and tau P301L/S320F. This suggests that the effect of Hsp22 NTDΔ is not specific to an aggregation prone form of tau, as tau WT did not result in appreciable aggregation in our cell model. However, reduction of tau P301L/S320F levels by Hsp22 NTDΔ did result in a concomitant decrease in aggregated tau found in the insoluble fraction. This demonstrates that targeting normal soluble tau for clearance could be an adequate strategy for therapy, without the need for specifically targeting aberrant or aggregating forms. This is consistent with previous work demonstrating that when tau expression was turned off in an animal model of tau aggregation, memory function recovered and neuronal death was inhibited [47]. If a single therapeutic molecular chaperone construct targets many variant species of tau found in neurodegenerative disease (whether these be disease-relevant mutations, post-translationally modified forms, or specific oligomeric or fibrillar species), there would be no need to stratify patients with a personalized molecular characterization of each individual presenting tauopathy.

## 4. Materials and Methods

### 4.1. Hsp22 mRNA Expression in Human Brain Tissue

Microarray postmortem tissue for mRNA analysis was obtained from 7 different ADR brain banks by UC Irvine Brain Bank (Irvine, CA, USA). Microarray analysis was undertaken on 4 brain regions (PCG, SFG, HPC and EC) from AD cases (*n* = 26, age 74–95, mean age 85.7 ± 6.5 yrs), age-matched controls (*n* = 33, age 60–99 yrs, average age 84.2 ± 6.5 yrs) and young normal cases (ages 20 to 59; *n* = 22, mean 35.4 ± 10.5 years) using Affymetrix (Santa Clara, CA, USA) arrays 9HgU133 plus 2.0 as described previously [48]. All subjects were evaluated to be cognitively normal or AD based on diagnoses from medical records. mRNA expression of each sample was evaluated by gene-chip hybridization as described by Cribbs et al. [48]. These data are deposited on the MIAME-compliant GEO database (Accession number GSE11882).

### 4.2. Transgenic Animal Brain Tissue Preparation

Four rTg4510 mice (Jackson Laboratories, Bar Harbor, ME, USA) and four nontransgenic littermates were euthanized at 9 months by Somnasol overdose followed by transcardial perfusion with 0.9% saline. The left hemisphere was obtained from each mouse and fixed overnight in 4% paraformaldehyde followed by cryo-protection in sucrose gradients of 10%, 20% and 30%. Brains were frozen on a freezing stage and horizontally sectioned on a sliding microtome at 25 µm then stored in a PBS solution containing 0.02% NaN_3_ at 4 °C. All animal studies were approved by the University of South Florida Institutional Animal Care and Use Committee and carried out in accordance with the National Institutes of Health (NIH) guidelines for the care and use of laboratory animals (Aproval Code M1309, approved July 15, 2015).

### 4.3. Anti-Hsp22 Immunohistochemistry Analysis

Human tissue from the UC Irvine Brain Bank (Irvine, CA, USA) or mouse tissue was stained using free-floating technique. Sections were briefly incubated in PBS with 10% MeOH and 3% H_2_O_2_ to block endogenous peroxidases followed by three PBS washes. Sections were then permeabilized for 30 min in 0.2% Triton X-100 with 1.83% lysine and 4% goat serum (Lampire Biological Laboratories, Ottsville, PA, USA) in PBS. Sections were incubated in Hsp22 primary antibody (StressMarq, SPC-181D 1:100; Victoria, BC, Canada) at room temperature overnight. After three PBS washes, sections were incubated at room temperature in biotinylated goat anti-rabbit (Southern Biotech, 1:1,000; Birmingham, AL, USA) secondary antibody for 2 h. A Vectastain ABC kit (Vector Laboratories; Burlingame, CA, USA) was used to amplify visibility. After two PBS washes and one TBS wash, sections were incubated with 0.05% diaminobenzidine and 0.5% nickel for five minutes then developed with 0.03% H_2_O_2_. Stained sections were then mounted on charged slides and allowed to air-dry overnight followed by dehydration in EtOH gradients. Slides were cleared with Histoclear (National Diagnostics; Altanta, GA, USA) then coverslipped with distyrene, plasticizer, and xylene (DPX).

### 4.4. Analysis of Intrinsic Disorder Predisposition

Functional disorder predisposition profiles of human Hsp22 (UniProt ID: Q9UJY1) were evaluated by the D^2^P^2^ platform available at: http://d2p2.pro [49]. Per-residue disorder predispositions were also evaluated by a disorder meta-predictor, PONDR^®^ FIT [50], which was also used to study the effects of S24A/S57A, S24D/S57D, and NTDΔ mutations on the propensity of human Hsp22 for intrinsic disorder from amino acid sequence. PONDR^®^ FIT uses outputs from PONDR^®^ VLXT, PONDR^®^ VSL2, PONDR^®^ VL3, FoldIndex, IUPred, and TopIDP and shows noticeably improved prediction accuracy compared to single component predictors.

### 4.5. Molecular Cloning

Tau WT, tau P301L, tau ΔK280, Hsp22 WT, Hsp22 S/D, Hsp22 S/A and Hsp22 NTDΔ were subcloned into the multiple cloning site of a pET28a vector (Millipore Sigma #69864; Burlington, MA, USA) modified with an additional tobacco etch virus protease cleavage site (pET28a-TEV). Mutations in Hsp22 were constructed by site-directed mutagenesis or deletion mutagenesis utilizing the Q5 high fidelity DNA polymerase. Primers were designed using NEBaseChanger online tools (New England Biolabs, Ipswich, MA, USA). Correct mutagenic constructs were confirmed by sanger sequencing. For mammalian expression studies, tau cDNA constructs were used in pRK5 (BD Biosciences #556104; San Jose, CA, USA) vectors and Hsp22 cDNA constructs were used in pCMV6-XL6 (Origene #PCMV6XL6; Rockville MD, USA) vectors.

### 4.6. Protein Expression and Purification

One Shot BL21 Star (DE3) Chemically competent E. coli BL21 cells (ThermoFisher, Waltham, MA, USA) were transformed with 4R0N tau WT, 4R0N tau P301L, 4R0N tau ΔK280, Hsp22 WT, Hsp22 S/A, Hsp22 S/D or Hsp22 NTDΔ plasmids in a pET28a-TEV vector. The cells were then grown at 37 °C in LB media containing 50 μg/mL kanamycin. Once their OD600 reached 0.8 the cells were induced with 1mM of IPTG for 3 h. Centrifugation at 3320 *g* for 20 min was used to harvest the cells, which were then resuspended with 35 mL nickel chromatography running buffer (20 mM Tris-HCl pH 8.0, 500 mM NaCl, 10 mM Imidazole) containing EDTA-free protease inhibitors. The cells were then lysed using a freeze-thaw cycle followed by sonication. The lysed cells were centrifuged at 50,000 *g* for 30 min at 4 °C. The supernatant was affinity purified using a standard gravity column packed with 13 mL HisPur™ Ni-NTA Resin (Fisher Scientific, Waltham, MA, USA). After a wash with nickel chromatography running buffer, the protein of interest is eluted with 25 mL of elution buffer (20 mM Tris-HCl pH 8.0, 500 mM NaCl, 250 mM Imidazole). The eluted fractions were treated with 1 mL of TEV protease (2 mg/mL) for 5 h at room temperature then dialyzed back into nickel chromatography running buffer (1 L) overnight. A second nickel purification column was run wherein the flow through fractions containing the His Tag-free proteins were collected and the efficiency of the TEV cleavage was assessed by SDS-PAGE followed by Coomassie staining. For the tau constructs, size exclusion chromatography was performed using a HiLoad 16/600 Superdex 200pg column (GE Healthcare; Chicago, IL, USA) with a Bio-Rad NGC QUEST 10 Chromatography system (Bio-Rad, Hercules, CA, USA) in SEC buffer (20 mM Tris-HCl pH 7.6, 500 mM NaCl, 0.5mM EDTA, 0.5 mM DTT). Fractions of interest were pooled and concentrated. The concentrated protein was then aliquoted, flash frozen with liquid nitrogen, and frozen at −80 °C until use.

### 4.7. Thioflavin T Fluorescence Assay

Recombinant tau and Hsp22 proteins were dialyzed into 100 mM sodium acetate buffer pH 7, overnight. Then, 10 µM of tau P301L, 10 µM of tau ΔK280, or 20 µM of tau WT was mixed with substoichiometric concentrations of WT Hsp22, Hsp22 S/A, Hsp22 S/D, or Hsp22 NTDΔ as indicated, as well as 10 µM heparin and 10 µM thioflavin T. A total of 100 µL of each assay sample was loaded onto 96-well black clear-bottom plates (Fisher Scientific, Cat#07-200-525; Hampton, NJ, USA) in triplicate. Fluorescence was then measured with 440nm excitation and 482 nm emission using a Cytation 3 multi-mode reader (BioTek; Winooski, VT, USA). Tau was induced to aggregate with 10 µM heparin and aggregation was followed over a 72-hour period taking readings every 10 min.

### 4.8. Transmission Electron Microscopy

Ten µL of protein samples at the end of the thioflavin T assay were adsorbed onto prewashed 200 mesh formvar/carbon-coated copper grids for 5 min. The grids were washed with water (10 μL) two times, stained with 10 μL filtered 2% uranyl acetate for 1 min, then dried. The samples were analyzed with a JEOL (Peabody, MA, USA) 1400 Digital Transmission Electron Microscope, and images were captured with a Gatan (Pleasanton, CA, USA) Orius wide-field camera at the Electron Microscopy Core Facility in the College of Medicine at the University of South Florida. The fields shown are representative.

### 4.9. Differential Scanning Fluorimetry

To assess protein thermal stability, Hsp22 WT and Hsp22 NTDΔ were diluted to a final concentration of 5–10 μM in 100 mM sodium acetate buffer pH 7.0 containing 2 mM DTT and 5× Sypro Orange dye (Invitrogen). The 30 μL mixtures were dispensed into a Bio-Rad 96-well thin-wall PCR plate and sealed with microplate adhesive film. Sypro Orange fluorescence was monitored as a function of temperature in a Bio-Rad (Hercules, CA, USA) CFX96 Touch Real-Time PCR Detection System through use of the C1000 Touch Thermal Cycler and FRET channel. Thermal melts were conducted in triplicate for each condition by heating from 25 to 95 °C in 1 °C increments for 2 min and measuring fluorescence at each temperature step. Fluorescence data were blank-subtracted, normalized to the maximum signal for each individual melt, and Boltzmann Sigmoid analysis was conducted using GraphPad Prism to determine the melting temperature (Tm) from two independent experiments.

### 4.10. HEK 293T Co-Transfection and Fractionation

HEK293T cells (ATCC; Old Town Manassas, VA, USA) were subcultured in 6-well plates coated with poly-L-Lysine. At ~60–80% confluence, cells were co-transfected using polyethylenimine with tau and Hsp22 expression plasmids at a ratio of 1:5 and 1:10, respectively. Forty-eight hours after transfection, cells were harvested and lysed in 400 µL of Triton X-100 lysis buffer (150 mM NaCl, 50 mM Tris, 1 mM EDTA, 1% Triton X-100) supplemented with protease inhibitor cocktail, phosphatase inhibitor cocktails and PMSF (Sigma, P8340, P0044, P5726 and P7626; St. Louis, MO, USA). After a 20-minute incubation on ice, samples were cleared by centrifugation at 10,000× *g* for 10 min. Samples were then centrifuged at 100,000× *g* for 30 min to pellet triton-insoluble proteins. Supernatant was collected as the triton-soluble fraction. Pellets were washed once in 1 mL of lysis buffer, centrifuged again at 100,000× *g* for 30 min, resuspended in 8M urea and used as the triton-insoluble fraction.

### 4.11. Western Blots

Triton-soluble and -insoluble fractions were diluted into 1X SDS sample buffer. Triton-soluble samples were incubated at 99 °C for 3 min and all samples were processed by SDS-PAGE in precast Any Kd gradient gels (BioRad; Hercules, CA, USA) and transferred to PVDF membranes. PVDF membranes were reversibly stained with Ponceau S to confirm proper transfer and equivalent loading of total protein in each lane. Rabbit anti-tau polyclonal antibody (Dako/Agilent, A0024; Santa Clara, CA, USA) was used to detect total tau protein. Goat anti-HspB8 (Hsp22) antibody (ThermoFisher, PA5-18103, directed to the CTD; Waltham, MA, USA) was used to detect Hsp22 protein.

### 4.12. Statistical Analysis

Quantified data are expressed as mean ± standard error of the mean (SEM) for at least three independent experiments. Significant differences relative to controls were evaluated using a one-way analysis of variance (ANOVA) followed by either Dunnet’s or Fisher’s LSD post hoc test using Prism software (Graphpad Software, Inc. La Jolla, CA, USA).

## Figures and Tables

**Figure 1 ijms-21-05442-f001:**
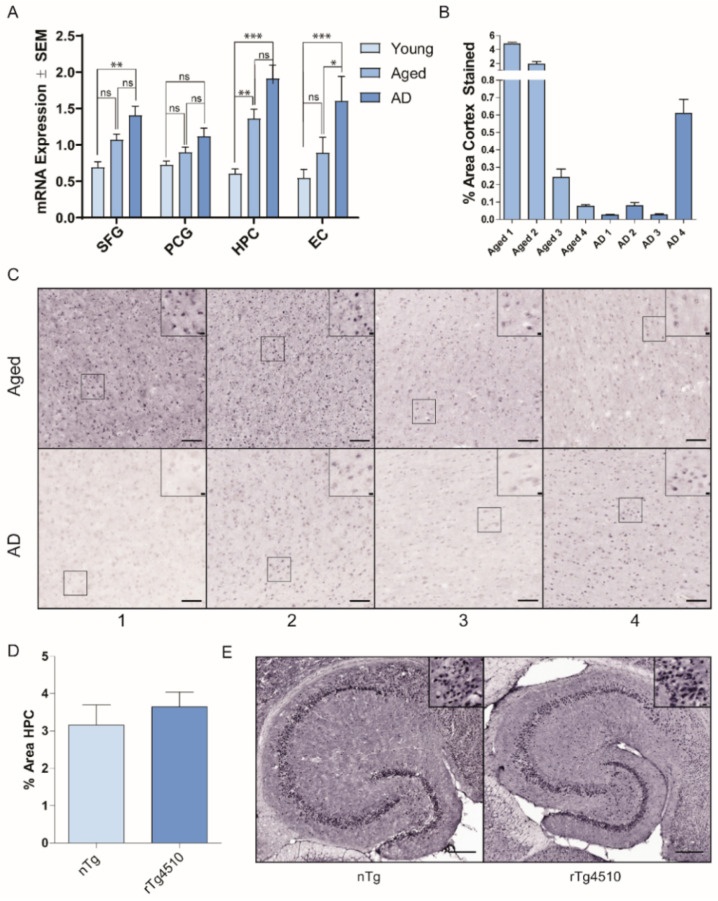
Expression levels of Hsp22 at the mRNA and protein levels. (**A**) Microarray analysis was undertaken on 4 human brain regions, the superior frontal gyrus (SFG), posterior cingulate gyrus (PCG), hippocampus (HPC) and entorhinal cortex (EC), from young normal cases (ages 20 to 59; *n* = 21,18,19,21, respectively), age-matched controls (age 60–99 yrs, *n* = 14, 21, 21, 21, respectively), and AD cases (age 74–95, *n* = 15, 18, 24, 21, respectively). Data were analyzed by one-way ANOVA with Dunnet’s post hoc comparisons test, * *p* < 0.05; ** *p* < 0.01; *** *p* < 0.001). (**B**,**C**) Medial temporal gyrus tissue samples from four aged and four AD brains were evaluated by immunohistochemical staining with an antibody to Hsp22. The mean area of medial temporal gyrus Hsp22 positive staining from 10 random fields per patient is shown. Representative images from each patient are shown. Scale bars reflect 100 µm and 10 µm (inset boxes). (**D**) Mean % area of Hsp22 staining in the hippocampus from 4 non-transgenic and 4 rTg4510 mice. (**E**) Representative images of Hsp22 staining in nTg and rTg4510 brain tissue. Scale bars reflect 200 µm and 10 µm (inset boxes).

**Figure 2 ijms-21-05442-f002:**
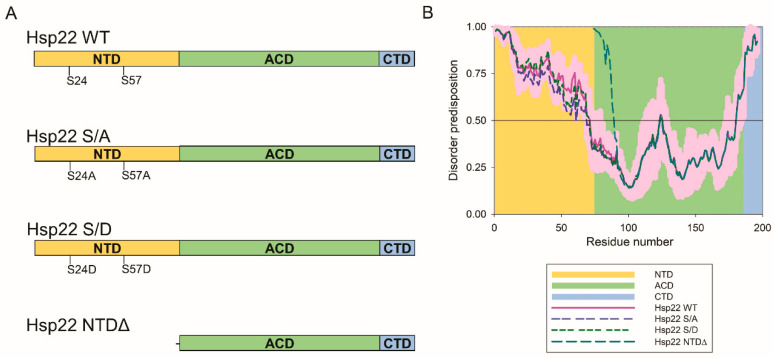
Hsp22 variants and intrinsic disorder propensity profiles. (**A**) Schematics of the four Hsp22 variants examined show the NTD (N-terminal), ACD (α-crystallin domain) and CTD (C-terminal) regions as well as the sites of point mutations. (**B**) Disorder predisposition profiles using PONDR^®^ FIT are shown for the four Hsp22 variants examined in this study. Light pink shadows around the curves show error distributions for Hsp22 WT. The background shading corresponds to amino acid residues within each domain as indicated.

**Figure 3 ijms-21-05442-f003:**
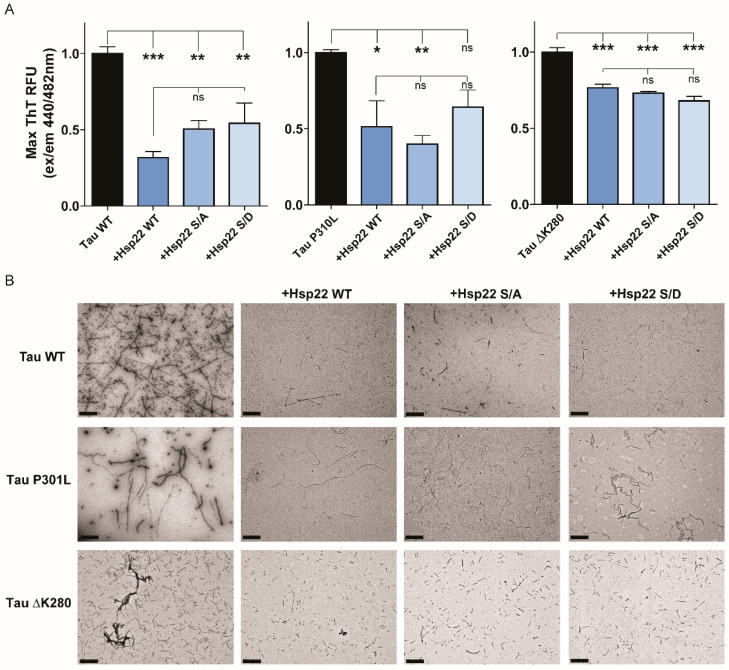
Hsp22 prevents tau aggregation in vitro. (**A**) WT, P301L and ΔK280 tau aggregation was monitored by Thioflavin T (ThT) fluorescence intensity over 72 h. Bars represent the mean maximum ThT intensity (± SEM) normalized to the no chaperone protein condition (black bars). Hsp22 WT (dark blue), Hsp22 S/A (S24A/S57A, medium blue), and Hsp22 S/D (S24D/S57D, light blue) were evaluated at a 20:1 tau:chaperone ratio. Data were analyzed by one-way ANOVA with Tukey’s post hoc multiple comparisons test (*n* = 6, two independent experiments performed in triplicate), * *p* < 0.05; ** *p* < 0.01; *** *p* < 0.001). (**B**) Representative 20,000× transmission electron microscopy images of tau alone or in the presence of the indicated Hsp22 variant. Scale bar 1 µm.

**Figure 4 ijms-21-05442-f004:**
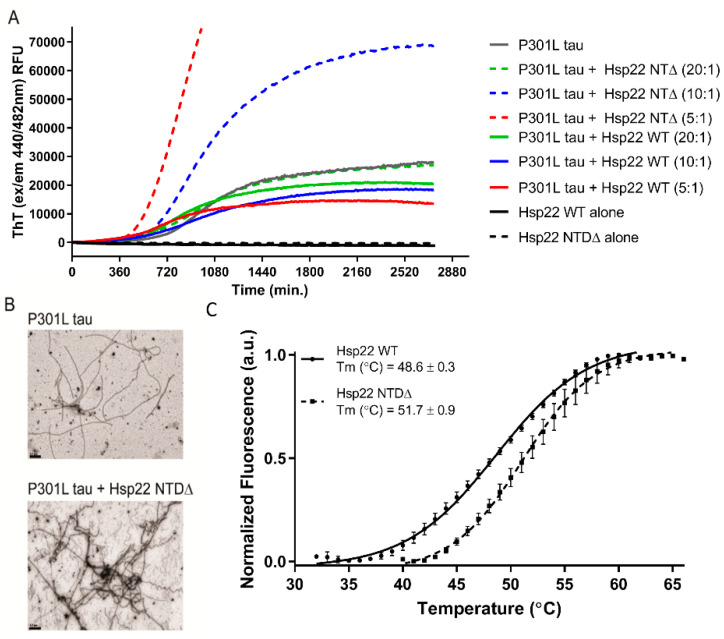
Hsp22 NTDΔ accentuated aggregation of tau P301L in a concentration-dependent manner. (**A**) Tau P301L (2.5 µM) aggregation was monitored by ThT fluorescence intensity over 40 h in the presence of either Hsp22 WT or Hsp22 NTDΔ at tau:chaperone ratios of 20:1, 10:1 or 5:1. (**B**) Representative 20,000× transmission electron microscopy images of tau P301L alone or in the presence of the Hsp22 NTDΔ variant. Scale bar 0.5 µM. (**C**) Differential scanning fluorimetry assessment of melting temperatures (Tm) for purified recombinant Hsp22 WT and Hsp 22 NTDΔ; a.u. = arbitrary units of fluorescence.

**Figure 5 ijms-21-05442-f005:**
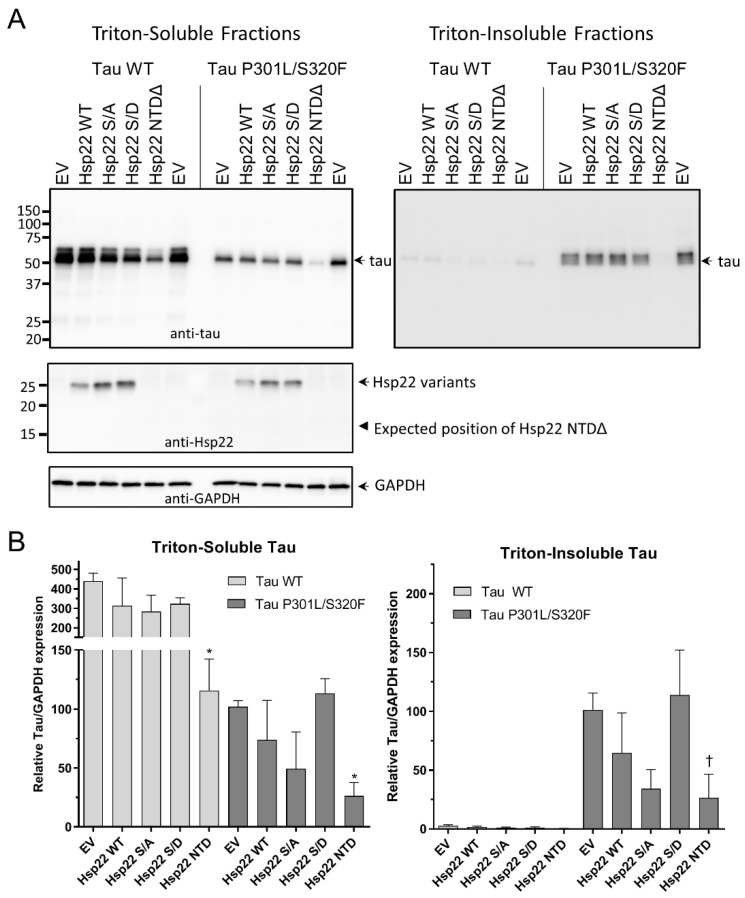
Changes in cellular expression of tau WT or tau P301L/S320F with co-expression of Hsp22 and variants at a 1:5 tau vector: Hsp22 vector ratio. (**A**) Representative Western blot images of triton-soluble and triton-insoluble fractions from HEK293T cells co-transfected with tau WT or tau P301L/S320F and either an empty vector plasmid (EV) or one of the variant Hsp22 expression vectors as indicated. Immunoblots were with antibodies specific for tau, Hsp22 or GAPDH as indicated. (**B**) Quantitation of tau blots from three independent experiments, tau/GAPDH ratios were normalized to the level of tau expression in the tau P301L/S320F with empty vector co-transfection sample. Data were analyzed by one-way ANOVA with Dunnet’s post hoc comparisons test to each empty vector control (*n* = 3, * *p* < 0.05; ^†^ indicates *p* > 0.05 with Dunnet’s and *p* < 0.05 with Fisher’s LSD).

**Figure 6 ijms-21-05442-f006:**
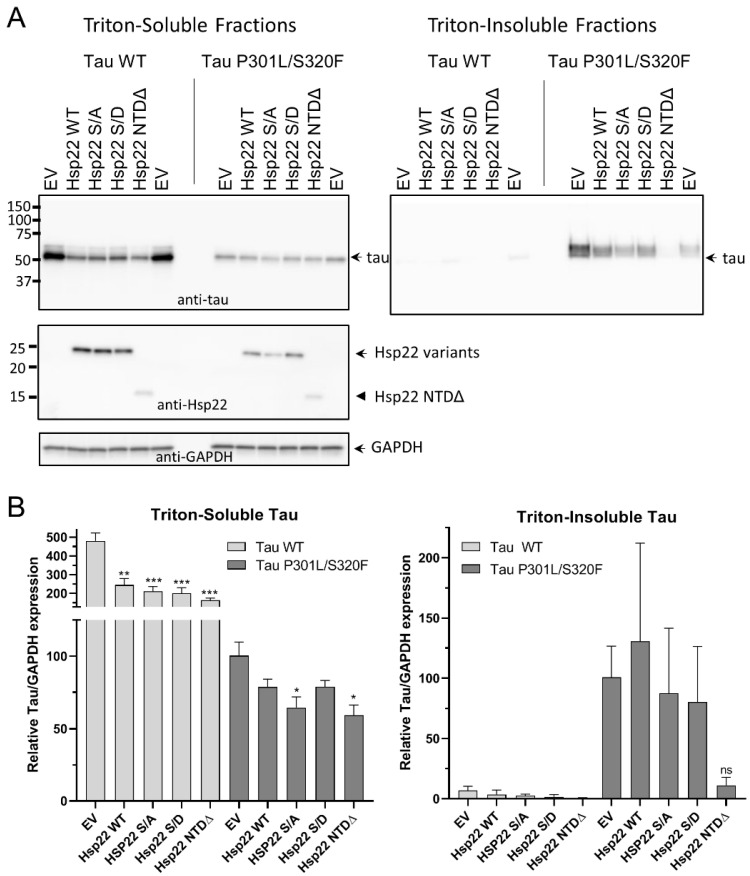
Changes in cellular expression of tau WT or tau P301L/S320F with co-expression of Hsp22 and variants at a 1:10 tau vector: Hsp22 vector ratio. (**A**) Representative Western blot images of triton-soluble and triton-insoluble fractions from HEK293T cells co-transfected with tau WT or tau P301L/S320F and either an empty vector plasmid (EV) or one of the variant Hsp22 expression vectors as indicated. Immunoblots were with antibodies specific for tau, Hsp22 or GAPDH as indicated. (**B**) Quantitation of tau blots from three independent experiments, tau/GAPDH ratios were normalized to the level of tau expression in the tau P301L/S320F with empty vector co-transfection sample. Data were analyzed by one-way ANOVA with Dunnet’s post hoc comparisons test to each empty vector control (*n* = 3, * *p* < 0.05; ** *p* < 0.01; *** *p* < 0.001; ns = not significant).

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
