# Peer review of "Hsp22 with an N-Terminal Domain Truncation Mediates a Reduction in Tau Protein Levels"

_ijms, 2020, doi:10.3390/ijms21155442_

Round 1
Reviewer 1 Report
The revised manuscript has addressed my concerns.
There are a few minor corrections the authors should make before publication:
1. On line 317, "Tau protein levels...." should more specifically state "Levels of tau P301L/S320F..."
2. The legend to Figure S3 refers to a Western blot in Figure 4, but there is no blot in Figure 4. I believe the authors meant to refer to Figure 5 instead.
3. The legend to Figure S4 states the Western blot detects relative levels of HspB8, which is apparently another name for Hsp22. To avoid confusion and to maintain consistency with the rest of the paper, I suggest replacing HspB8 with Hsp22 in the legend or using both names (HspB8/Hsp22).
4. The supplementary data is variously referred to as Supplementary Figure S1 (line 220), Supplemental Figure S2 or S4 (lines 258 and 311-312), or Figure S3 (line 293). I suggest using a single standard description for the supplementary data.
Author Response
- On line 317, "Tau protein levels...." should more specifically state "Levels of tau P301L/S320F..."
Thank you, this sentence has been changed to “Tau P301L/S320F protein levels….” [New line 231]
- The legend to Figure S3 refers to a Western blot in Figure 4, but there is no blot in Figure 4. I believe the authors meant to refer to Figure 5 instead.
Thank you for pointing out this error. Yes, we meant to refer to Figure 5 and have changed the text in the legend to Figure S3 to reflect this.
- The legend to Figure S4 states the Western blot detects relative levels of HspB8, which is apparently another name for Hsp22. To avoid confusion and to maintain consistency with the rest of the paper, I suggest replacing HspB8 with Hsp22 in the legend or using both names (HspB8/Hsp22).
Thank You, Yes HSPB8 is the gene name and HspB8 is often an alias for the Hsp22. We changed the legend and figure to Figure S4 to remove HspB8 and replace with Hsp22 for consistency and clarity.
- The supplementary data is variously referred to as Supplementary Figure S1 (line 220), Supplemental Figure S2 or S4 (lines 258 and 311-312), or Figure S3 (line 293). I suggest using a single standard description for the supplementary data.
Thank You, we changed all instances referring to Supplementary Figures to a consistent format of “Supplementary Figure S#”. This has been made consistent in each instance you identified. Line 220 [New line 171], line 258 [New line 201], lines 311-312 [New line 226], and line 293 [New line 222]. Additional instances at new lines 204 and 352 were also changed for consistency.
Reviewer 2 Report
The authors have answered my question why they used three different variants of tau protein in their experiments. I have no further comments about the revised manuscript. As I wrote before the paper is well written and presented and I find this contribution important for the field neurodegeneration studies. Therefore I would like to recommend its publication in IJMS.
Author Response
No revision requested.
Reviewer 3 Report
Hsp22 with an N-terminal domain truncation mediates a reduction in tau protein levels
By Webster, J.M. et al.
This is a revised version of the manuscript with the same title. The authors conducted additional experiments and modified the text according to the comments of the reviewers. The revision has cleared most of the previous concerns, and the manuscript is much better and convincing than the original. The negative control experiment for Figure 3 is still missing, but now I understand that it would be hard to find an appropriate non-relevant control protein for repeating the experiment. I believe this manuscript is now suitable for publication in International Journal of Molecular Sciences with a few minor modifications.
- Page 8, line 9 (line 317): “Tau protein levels were also reduced in the presence of all the Hsp22 variants ...” -> Tau P301L/S320F protein levels ?
- Figure 6 and Page 10, line 3 (line 408): “... resulted in a significant reduction in tau WT levels for all of the Hsp22 variants examined.” -> Do the authors suggest that the effect of Hsp22 (WT, S/A, S/D) is also the result of the enhancement of tau clearance in cells? This point should be described clearly.
//End.
Author Response
- Page 8, line 9 (line 317): “Tau protein levels were also reduced in the presence of all the Hsp22 variants ...” -> Tau P301L/S320F protein levels ?
Thank you, this sentence has been changed to “Tau P301L/S320F protein levels….” [New line 231]
- Figure 6 and Page 10, line 3 (line 408): “... resulted in a significant reduction in tau WT levels for all of the Hsp22 variants examined.” -> Do the authors suggest that the effect of Hsp22 (WT, S/A, S/D) is also the result of the enhancement of tau clearance in cells? This point should be described clearly.
Thank you for facilitating a clearer description. We demonstrated that Hsp22 (WT, S/A, S/D) overexpression does result in a reduction in tau WT protein levels at the 1:10 tau:sHsp ratio. But only Hsp22 NTD delta reduces tau levels at the lower 1:5 ratio. We now describe this more explicitly in the discussion. The text [New lines 297-300] now reads… “While only Hsp22 NTDΔ significantly reduced tau levels in our standard experiment using plasmid vectors at a 1:5 ratio of tau to sHsp; further increasing the ratio of sHsp to tau vectors resulted in a significant reduction in tau WT levels for all of the Hsp22 variants examined. Overall, this suggests that Hsp22 WT, Hsp22 S/A Hsp22 S/D, and Hsp22 NTDΔ are each capable of enhancing the clearance of tau in cells, however, Hsp22 NTDΔ has a more potent effect on cellular tau clearance. This suggests a potential regulatory role of the Hsp22 NTD toward the client tau that is independent of phosphorylation at serines 24 and 57.”
Additional Edits:
a misspelling typo was corrected on new line 229 … “performed”
a misspelling typo was corrected on new line 60…”pseudophosphorylated”
a misspelling typo was corrected on new line 80…”non-phosphorylatable”
This manuscript is a resubmission of an earlier submission. The following is a list of the peer review reports and author responses from that submission.
Round 1
Reviewer 1 Report
Members of the small heat shock protein (Hsp) family can disrupt the aggregation of amyloidogenic clients. This manuscript examines the effects of Hsp22 on tau aggregation. Using ThT and TEM, the authors found Hsp22 could partially inhibit tau aggregation at a 20:1 ratio of tau:Hsp22. The phosphorylation status of Hsp22 did not substantially alter Hsp22 chaperone activity against tau. An N-terminal deletion of Hsp22 (NTDΔ) did not alter tau aggregation at a 20:1 ratio of tau:Hsp22 but accelerated tau aggregation at higher relative concentrations of Hsp22. Hsp22 NTDΔ lowered tau expression levels in co-transfection experiments, whereas wild-type Hsp22 had no effect. Examination of Hsp22 in brain samples found elevated mRNA but relatively low protein levels in aged and neurodegenerative brains.
The experiments are solid, the data interpretation is sound, and the manuscript is well-written. The overall work does not support a major neuroprotective role for Hsp22 in Alzheimer's Disease, but this information is still valuable to the field. As described below, the manuscript could be strengthened by providing additional background information, further insight into the experimental design, and more comprehensive data interpretation. The only major weakness is a missing control for the co-transfection experiment that could greatly aid interpretation of the data.
Major point
The NTDΔ variant of Hsp22 enhanced tau aggregation in vitro but greatly reduced tau expression in a co-transfection experiment. Expression of Hsp22 NTDΔ itself was also greatly reduced in co-transfected cells. The authors suggest that Hsp22 NTDΔ may contribute as a constituent to the aggregation of tau and may work with other proteins to clear tau from cells. However, it is also possible that Hsp22 NTDΔ expression - either alone or in combination with tau - is toxic and consequently reduces protein expression. To address this possibility, the authors should compare the viability of their co-transfected cells to the viability of mock transfected cells, cells transfected with tau alone, and cells transfected with Hsp22 NTDΔ alone.
Cells transfected with Hsp22 NTDΔ alone represent an important control. This missing control could determine whether Hsp22 NTDΔ expression is toxic and would also determine whether Hsp22 NTDΔ protein levels are low when it is expressed in the absence of tau. If Hsp22 NTDΔ is incorporated into a tau aggregate that is subsequently degraded, then transfection of Hsp22 NTDΔ alone should produce a much higher level of Hsp22 NTDΔ protein than seen for its co-transfection with tau. If transfection of Hsp22 NTDΔ alone still results in low Hsp22 NTDΔ expression levels, it is possible the protein is toxic - which would be confirmed from the viability assays suggested above.
These simple experiments would provide additional data for understanding the potential mechanisms of Hsp22 NTDΔ action as outlined in the Discussion.
Minor points
A more comprehensive analysis of the data in Figure 4 would be appropriate for both the Results and Discussion. Increasing concentrations of the Hsp22 NTDΔ construct did not just increase tau aggregation - they both reduced the lag phase and accelerated the kinetics fibrillization. These anti-chaperone effects are consistent with the possible direct contribution of Hsp22 NTDΔ to aggregate formation that was considered in the Discussion. The lag phase does not appear to be substantially affected by wild-type Hsp22. The inhibitory effect instead seems to function at mid- to late-stage aggregation, and there is only a minor concentration-dependent effect on inhibition. What is the proposed mechanism for wild-type Hsp22 inhibition of tau aggregation?
The Introduction discusses the inhibitory effects of Hsp27 on tau aggregation. It would be interesting to know if these effects are similar to what was observed with Hsp22 by ThT and TEM. Like Hsp22, did Hsp27 have no inhibitory effect on the lag phase of aggregation? Were tau fibrils reduced and thinner in the presence of Hsp27, which was seen for Hsp22?
p7 lines 209-210 states the NTD of small Hsps is often a flexible disordered region. This would be good information to provide as a rationale for Figure 2b (ie, mention this in the Results as well as the Discussion).
The Legends for Figures 2 and 3 use capital letters in reference to the Figure panels, whereas the other Figure legends and Figures themselves use lower case letters.
The Figures are well-organized, but Figures 5 and 6 could possibly be enlarged to improve the legibility of text within the Figures.
p4 line 110: There are two periods at the end of the sentence.
p7 lines 213-214: Three constructs are mentioned, but the following sentence refers to them as "both".

Reviewer 2 Report
The manuscript presents in vitro and in vivo studies of the effect of small heat shock protein Hsp22 on tau protein aggregation. The authors studied wild type Hsp22, a non-phosphorylatable (S24A/S57A) and pseudo-phosphorylated (S24D/S57D) mutants and N-terminal domain deletion mutant. First of all the authors show that Hsp22 mRNA expression is upregulated in the human brain with aging and Alzheimer's disease, but the protein levels from the brain tissue did not correlate with the upregulation. In the discussion, they wrote that this suggests a compensatory role for Hsp22 in the brain. In vitro experiments show that Hsp22 and both S24A/S57A and S24D/S57D mutants reduce the aggregation of tau protein but the effect of the mutants is not statistically different than that Hsp22 protein. The authors should discuss in more detail why they used three different variants of tau protein (WT, P301L, and DeltaK280) in these experiments. In vitro experiments for Hsp22 N-terminal domain deletion mutant show an increase in tau P301L aggregation with the Hsp22 mutant concentration.
In vivo experiments show that tau expression is slightly reduced by coexpression of Hsp22 and Hsp22 S24A/S57A mutant but not by S24D/S57D mutant. Only Hsp22 N-terminal domain deletion mutant coexpression resulted in a significant reduction of tau protein level. Analysis of insoluble aggregates indicated that a reduction of aggregation cannot be directly attributed to an effect of the Hsp22 on tau aggregation but rather reduction in total tau protein concentration. This very different result of Hsp22 N-terminal domain deletion mutant on tau protein in vitro and in vivo is an interesting discovery. The paper is well written and presented and I find this contribution important for the field neurodegeneration studies. Therefore I would like to recommend its publication in IJMS.
Reviewer 3 Report
Hsp22 with a N-terminal domain truncation mediates a reduction in tau protein levels
By Jack Webster et al.
This manuscript deals with a role of a small heat shock protein/molecular chaperone Hsp22 on Tau protein aggregation and its expression. First, the authors found that Hsp22 mRNA expression was higher in brains of aged human and also of AD patients. On the other hand, protein levels of Hsp22 varied between aged patients and were relatively low in AD patients’ brains. The reason of this discrepancy remains unclear. Then the authors biochemically examined the effect of Hsp22 protein on the formation of Tau aggregation in vitro. The results showed that wild type as well as phosphomimetic (Ser24Asp/Ser57Asp) and phosphoincapable (Ser24Ala/Ser57Ala) mutants of Hsp22 were able to reduce heparin-induced aggregation of wild type Tau. The same effect was also observed on Tau P301L, but not significantly on Tau delta-K280. In addition, the authors confirmed the suppressive effect of Hsp22 on the Tau aggregation by transmission electron microscopy. The authors observed that in contrast to wild type Hsp22, an N-terminal-domain-truncated mutant (Hsp22 NTDdelta) enhanced the Tau P301L aggregation in vitro. The reason for the difference between wild type and the deletion mutant remains uncharacterized. The authors then showed using a cultured cell model that co-expression of Hsp22 NTDdelta (but not the wild type nor the phosphomimetic/phosphoincapable mutants) with wild type and P301L/S320F Tau reduced the protein level of Tau. In the case of Tau P301L/S320F, a significant portion of expressed protein existed as a Triton insoluble form, and Hsp22 NTDdelta (but not wild type and other mutants) also reduced the amount of Triton-insoluble Tau. The authors suggest that these data indicate that Hsp22 NTDdelta enhances the clearance of Tau protein in cultured cells. All of these results altogether indicate that Hsp22 may regulate the expression and aggregation of Tau protein, however, the physiological significance of the role of Hsp22 NTDdelta remains unclear.
This is the first report on the role of Hsp22 on Tau physiology. The results are of importance and interest not only for chaperone biologists but also for those working on Tau protein and Alzheimer’s disease. However, some important control experiments are missing and pivotal questions are not answered and not adequately discussed in the text.
The points to be clarified or modified are described below.
Major Points:
Fig 3: Often simply adding any proteins (i.e. BSA) in the solution may prevent formation of aggregation of another protein. A control experiment with non-relevant protein should be included. In addition, an SDS-PAGE analysis of the purified Tau and Hsp22 should be presented to show the purity of proteins used.
Levels of Hsp22 mRNA were shown to be higher in aged and AD, but this is not the case for the levels of Hsp22 protein. Why? The reason for this discrepancy is not adequately described in the text.
Hsp22 NTDdelta strongly enhanced the Tau aggregation. The authors should suggest possible molecular mechanisms of the enhancement in the text.
Hsp22 effectively prevented Tau aggregation in the in vitro assay, but this is not the case in the cellular model. This is clear because co-expression of Hsp22 did not increase the Triton-soluble form of Tau P301L/S320F. Why? The authors should describe the reason for this discrepancy in the text.
Figure 5: Expression of a non-natural protein can be toxic for cells, resulting in general inhibition of protein expression and cell damage. This could be the reason why the protein levels of Tau (WT and P301L/S320F mutant) and Hsp22 NTDdelta were quite low in the Hsp22 NTDdelta-expressing cells. The authors should include appropriate control experiments to exclude this possibility. In addition, a loading control with a non-relevant protein is essential.
Minor Points:
Page 1, Title: “with a N-terminal domain” -> with an N-terminal domain ?
Page 10, section 4.5: The elution condition for the His-tag protein purification should be described more precisely.
Figure 6 and the corresponding text: The authors should indicate a possible reason why Hsp22NTDdelta expression is so low as compared with wild type Hsp22.
Lastly, I deeply feel sad and sorry to know that Dr. Chad Dickey, whom I met several times in scientific meetings many years ago, has deceased so early in his splendid career.
//End.